# Scorpion envenomation in the state of São Paulo, Brazil: Spatiotemporal analysis of a growing public health concern

Alec Brian Lacerda[1]*, Camila Lorenz[1], Thiago Salomão De Azevedo[1,2], Denise Maria Cândido[3], Fan Hui Wen[3], Luciano José Eloy[4], Ana Aparecida Sanches Bersusa[5], Francisco Chiaravalloti Neto[1]

1 Faculdade de Saúde Pública–Universidade De São Paulo, São Paulo, Brazil, 2 Secretaria de Saúde do Município de Santa Bárbara d'Oeste–SP, Santa Bárbara d'Oeste, Brazil, 3 Instituto Butantan, São Paulo, Brazil, 4 Centro de Vigilância Epidemiológica "Prof. Alexandre Vranjac", São Paulo, Brazil, 5 Superintendência de Controle de Endemias do Estado de São Paulo, São Paulo, Brazil

* alec.lacerda@usp.br

**Data Availability Statement:** All relevant data are within the manuscript and its Supporting Information files.

## Abstract

Scorpion envenomation is a significant public health concern in São Paulo, Brazil, and its incidence and mortality have increased in recent decades. The present study analyzed documented scorpion envenomation notifications from 2008 to 2018 throughout the 645 municipalities of São Paulo. Annual incidence and mortality rates were calculated and stratified according to sex and age. The local empirical Bayesian method and Getis-Ord Gi* statistic were used to represent standardized incidence rates in the municipalities and to identify high- and low-risk agglomerates. The incidence rate of scorpion envenomation quintupled between 2008 and 2018. Overall, the risk was higher for man, and increased with age. Deaths due to envenomation, however, were concentrated almost entirely in children 0–9 years of age. Incidence maps showed that the risk of envenomation increased in almost all regions and municipalities of São Paulo throughout the study period. The highest incidence rates were found in the western, northwestern and northern regions of the state, in contrast to the São Paulo metropolitan area and southern and coastal regions. Hot spots were identified in the Presidente Prudente, Barretos, São José do Rio Preto, and Araçatuba regional health districts, which over time formed a single high-risk cluster. In spatial terms, however, deaths were randomly distributed. In this study, we identified areas and populations at risk of scorpion envenomation and associated–fatalities, which can be used to support decision-making by health services to reduce human contact with these arachnids and avoid fatalities, especially in children.

## Introduction

There are many venomous animals with toxins that are potentially harmful to humans among the world's diverse fauna. Envenomations that can cause severe injuries or sequelae, particularly those with socioeconomic and medical repercussions, are considered important from a

**Funding:** ABL received funding from the Conselho Nacional de Desenvolvimento Científico e Tecnológico – CNPq, number: 130489/2020-4. The funder had no role in study design, data collection and analysis, decision to publish, or preparation of the manuscript. FCN received funding from the Conselho Nacional de Desenvolvimento Científico e Tecnológico – CNPq, number: 306025/2019-1. The funder had no role in study design, data collection and analysis, decision to publish, or preparation of the manuscript.

**Competing interests:** The authors have declared that no competing interests exist.

public health perspective, as they can result in temporary or permanent injuries and even fatalities [1]. Scorpions can inflict such envenomations, resulting in a significant and emerging public health concern, especially in the Middle East, India, Mexico, and Brazil [2–4].

There are approximately 2,621 species of scorpion worldwide, belonging to 23 distinct families [5]. Of these, only 30 are considered harmful to humans, 29 of which belong to the Buthidae family. In South America, and particularly in Brazil, *Tityus* is the most medically relevant scorpion genus due to the clinical manifestations caused by envenomations in humans and the high incidence in recent years [2, 3, 6]. There are four epidemiologically significant species in Brazil: *Tityus serrulatus*, *T. bahiensis*, *T. obscurus*, and *T. stigmurus* [7, 8]. The high incidence and severity of, as well as the difficulty controlling scorpion stings, in some tropical countries, have made envenomations a public health concern. In these countries there are more than 1,200,000 envenomations annually, resulting in over 3,000 deaths [2].

From a clinical point of view, scorpion envenomations can be classified as mild, moderate, or severe [7, 9–12]. In mild cases, the toxins in the scorpion venom usually cause local effects, while systemic symptoms of autonomous nervous system imbalance, including nausea and vomiting, sweating, tachycardia, tachypnea, and mild hypertension occur in moderate cases. Severe envenomations are generally associated with cardiovascular, pulmonary, metabolic, and neurological complications, and can result in death, especially in children under 10 years of age [7, 9, 12–14]. According to the Brazilian Ministry of Health Guidelines, antivenom treatment is recommended in which the patient presents with signs and symptoms of systemic envenomation, which are classified as moderate or severe cases depending on the severity of the clinical manifestations [11, 14, 15].

Although the vast majority of scorpion envenomations in Brazil are considered mild with benign outcome [16], numerous severe cases occur, with some resulting in deaths. According to Reckziegel and Pinto (2014) [17], the highest rates of scorpion envenomations and deaths were observed in the northeast and southeast regions of Brazil, between 2000 and 2012. In the southeast region, the average annual incidence was 19.1 cases per 100,000 inhabitants, with the state of São Paulo (SP) accounting for 13.3% of all cases and 3.4% of the total envenomation-related deaths recorded in the country during this period.

In Brazil, envenomations due to scorpion stings are a significant public health concern because of their increasing incidence, which is related to the close proximity of scorpions to humans, as well the adaptation to their habitat. Monitoring and control of these arachnids are essential to eliminate, or at least minimize, the risk of death due to an envenomation [18]. For this reason, it is extremely important to characterize scorpion envenomations in terms of the people affected by the stings, as well as, spatiotemporal factors.

Few studies have addressed scorpion envenomation in geographic terms [19–21], despite the evident influence of the environment, climate, and human occupation [21–25]. Spatial analysis tools available now offer results that can strengthen decision-making strategies and guide efforts to target higher-risk areas in an effort to fill in this knowledge gap. As such, the present study aimed to describe the occurrence of scorpion envenomations in SP between 2008 and 2018 based on case data, and assesses the temporal and spatial distribution of these envenomations to identify areas at higher or lower risk, with the goal of assisting in the development of strategies with which to monitor and control these epidemiological events.

## Methods

### Type, period, population, and study area

This descriptive and ecological study utilizes secondary data on scorpion envenomations in SP between 2008 and 2018; the data unit considered during analysis was each municipality in the

state. SP is located in southeastern Brazil (Fig 1), and includes 645 municipalities, which are divided into 17 Regional Health Districts (RHDs). With 46,289,333 inhabitants, the population density of SP is 166.23 inhabitants/km$^2$; it has the second-highest Human Development Index (HDI) score of any Brazilian state (0.783) [26].

The state of SP has three primary climates. According to the Köppen-Ginger classification scheme, the western plateau of the state has a tropical climate (*Aw*) characterized by wet summers and dry winters. The higher altitude regions located in the Atlantic plateau and basaltic cuestas have a high-altitude tropical climate (*Cwa* and *Cwb*), whcih is characterized by hot summers and cold winters. The lowland coastal region has a humid tropical climate (*Af*), which is characterized by being hot and humid all year round. The annual rainfall varies between 1,600 mm on the south coast and 2,700 mm on the north coast. Finally, the peripheral depression has a subtropical climate (*Cfa*), which is characterized by the occurrence of well-distributed rainfall throughout the year, with hot summers and cold winters [27]. Although very little of the vegetation native to SP remains, largely due to the agriculture of sugar cane and pastures, the state's territory has three types of vegetation cover: mangroves are found in the coast; patches of savanna are found in the peripheral depression and on the eastern border of the western plateau; and the predominant vegetative throughout the rest of the state is the Atlantic Forest (Fig 1) [28].

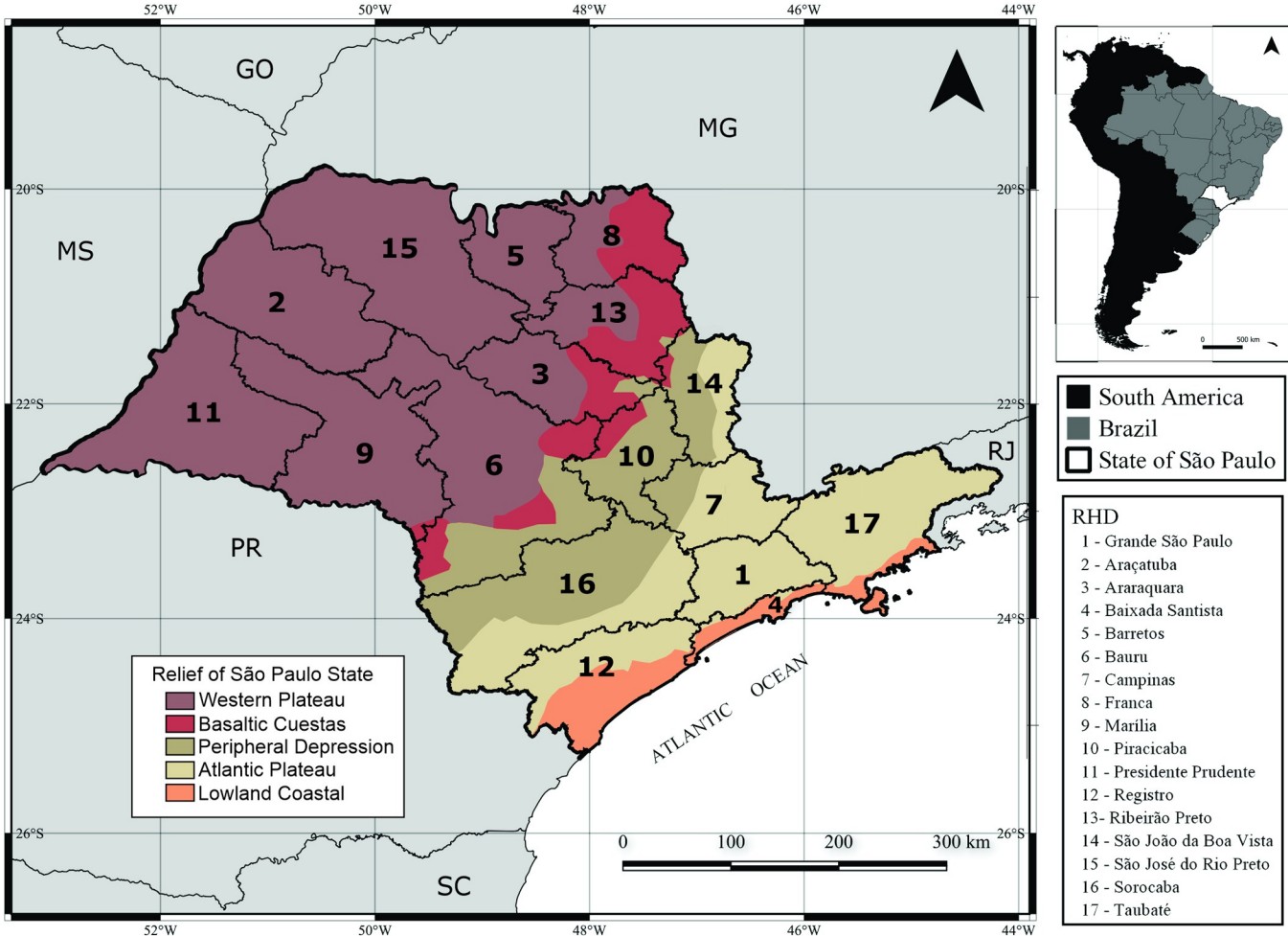

**Fig 1. Location of the state of São Paulo in Brazil and South America, its relief and the regional health departments (RHDs).**

## Data source, variables, and data analysis

Data regarding scorpion envenomations were obtained from the Brazilian Information System for Notifiable Diseases (SINAN) and were provided by the São Paulo State Epidemiological Surveillance Center (CVE). Geographic and demographic information on the municipalities was obtained from the Brazilian Institute of Geography and Statistics (IBGE), including shape maps. This information was entered into a database, and is also available in S1 File.

We calculated the incidence and mortality rates of scorpion envenomations in SP throughout the study period, taking age and sex into consideration. These rates were calculated annually and standardized by age and sex using the population of SP in 2013 as the standard. Linear and quadratic regression models were used to analyze the temporal series, and the rates were modelled per calendar year as a function of the year and the year raised to the square, respectively. We used the Shapiro-Wilk (SW) test to assess the normality of the residuals, and the Durbin-Watson (DW) test to assess the existence of temporal autocorrelation, for which DW = 0 was considered to correspond to the existence of positive autocorrelation, DW = 2 to no autocorrelation, and DW = 4 to negative autocorrelation [29]. Incidence rates for scorpion envenomations were calculated annually for the municipalities of SP and standardized by age and sex. We mapped these rates using the local empirical Bayesian method with the queen criteria for neighborhood contiguity between municipalities. The entire population of SP from 2013 was considered the standard, and the results of these analyses are presented as choropleth maps, which also include data regarding municipalities with deaths.

We utilized the Getis-Ord Gi* spatial analysis of the calculated incidence rate for the municipalities of SP to identify clusters with high and low risks of scorpion envenomations for each year of the study. This technique considers a neighborhood matrix between municipalities, and identifies local spatial associations. The results were allocated to each municipality to develop statistical maps. In the Gi* statistics, high values indicate areas of increased occurrences of the event/phenomenon, and low values indicate groupings of decreased occurrences [30]. Finally, we used the false discovery rate (FDR) to avoid the problem of multiple comparisons in local statistics; this method can prevent false positive clusters, and more efficiently filter the real spatial agglomerates of interest [31].

## Ethical considerations

The present study was developed using secondary data provided by the CVE (Secretary of Health of the state of São Paulo). The data form which was anonymized without names or addresses, and scorpion accidents were aggregated by municipality and year. The protocol for the present study was submitted for approval by the institutional ethics review board of the University of São Paulo School of Public Health (COEP FSP/USP, CAAE approval record 10457119.6.0000.5421, protocol number 3408558) and no consent was required because we used anonymized secondary data.

## Results

Between 2008 and 2018, there were 145,464 scorpion envenomations documented in the 645 municipalities of SP (Table 1). A 425% increase was observed from 2008 (5,788 cases) to 2018 (30,394 cases).

The incidence and mortality rates of scorpion envenomations were 33% and 86% higher in men than in women, respectively, throughout the study period. These rates were also higher in men of all age groups, with the exception of mortality in the young adult age groups. Incidence rates increased with age, at 68% higher for people aged 60 years and over than for children aged 0–9 years, while mortality exhibited a distinct trend, and was concentrated in children

**Table 1. Number of cases and deaths resulting from scorpion envenomation by age and sex and respective rates of incidence (100,000 inhabitant-years) and mortality (1,000,000 inhabitant-years) in the state of São Paulo, Brazil from 2008–2018.**

| Sex | Incidence and mortality rate | Age group | | | | | |
|---|---|---|---|---|---|---|---|
| | | 0-9 years | 10-19 years | 20-39 years | 40-59 years | 60+ years | Total |
| Male | Incidence | 19.3 | 26.2 | 33.8 | 42.5 | 52.9 | 34.7 |
| | (N) | 6605 | 10006 | 27024 | 24995 | 13433 | 82063 |
| | Mortality | 0.79 | 0 | 0 | 0.02 | 0.08 | 0.13 |
| | (N) | 27 | 0 | 0 | 1 | 2 | 30 |
| Female | Incidence | 16.7 | 24.3 | 25.2 | 29.6 | 32.4 | 26 |
| | (N) | 5481 | 8916 | 19907 | 18413 | 10684 | 63401 |
| | Mortality | 0.46 | 0.03 | 0.01 | 0.02 | 0 | 0.07 |
| | (N) | 15 | 1 | 1 | 1 | 0 | 18 |
| Total | Incidence | 18 | 25.3 | 29.5 | 35.8 | 41.3 | 30.3 |
| | (N) | 12086 | 18922 | 46931 | 43408 | 24117 | 145464 |
| | Mortality | 0.63 | 0.01 | 0.01 | 0.02 | 0.03 | 0.1 |
| | (N) | 42 | 1 | 1 | 2 | 2 | 48 |

younger than 9 years of age, with 42 of the 48 documented deaths (88%) occurring in this age group. These results demonstrate that although adults are more likely to be envenomated, children are at a higher risk of death from envenomation.

The incidence rate of scorpion envenomations quintupled in SP between 2008 and 2018, gradually increasing from approximately 14 cases per 100,000 inhabitants in 2008 to 70 in 2018 (Fig 2). Mortality from scorpion envenomation was constant in SP from 2008 through 2014, from which point it increased significantly, with approximately 0.3 deaths per 1 million inhabitants in 2018, a 244.2% increase in the period.

Table 2 presents the results of the quadratic regression models for incidence and mortality, which all coefficients (intercept, year, and year squared) were significant or at their limit (year, in the incidence model). Since the models used for incidence and mortality correspond to second-degree functions with a positive quadratic term coefficient, both incidence and mortality rates clearly showed a significant increase between 2008 and 2018.

We opted to use quadratic regression models because the linear models were not sufficient to adjust for temporal autocorrelation. For incidence, the DW value of the model, without

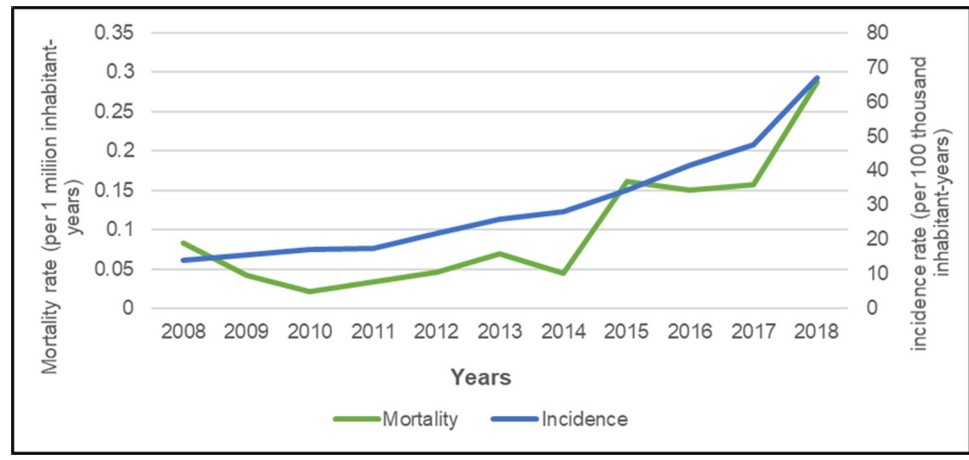

**Fig 2. Incidence and mortality rates of scorpion envenomations per year in the state of São Paulo, Brazil from 2008–2018.**

**Table 2. Ordinary least squares quadratic regression models for temporal modeling of incidence and mortality rates for scorpion envenomations in the state of São Paulo, Brazil from 2008 to 2018, and the results of the Durbin-Watson and Shapiro-Wilk tests.**

| Variables, tests and $R^2$ | Incidence rate | | Mortality rate | |
|---|---|---|---|---|
| | Coefficients/tests | p values | Coefficients/tests | p values |
| Intercept | 18.082 | 0.0004 | 0.103 | 0.0138 |
| Year | -2.647 | 0.0554 | -0.036 | 0.022 |
| Year$^2$ | 0.604 | 0.0002 | 0.005 | 0.002 |
| Durbin-Watson (DW) | 1.8098 | 0.1137 | 2.7858 | 0.7381 |
| Shapiro-Wilk (SW) | 0.9759 | 0.9388 | 0.9837 | 0.9832 |
| $R^2$ | 0.9766 | | 0.8918 | |

quadratic adjustment was 0.72, with $R^2 = 0.86$, whereas with the quadratic adjustment, the DW value was 1.80, $R^2 = 0.97$, showing improvement and adjustment of the model for the series autocorrelation. For mortality, the DW value without quadratic adjustment was 1.20, with $R^2 = 0.61$, while with adjustment the DW value was 2.78, $R^2 = 0.89$, also showing improvement and adjustment of the model for the series autocorrelation. The SW normality test for the residuals of the models for incidence and mortality did not reject the hypothesis of normal distribution in these time series.

The western, northwestern and northern regions of SP, which are part of the western plateau (Fig 1), had the highest numbers of scorpion envenomations during the study period (Figs 3 and 4). The highest incidence rates were found in the São José do Rio Preto, Barretos, Presidente Prudent and Araçatuba RHDs, although Araçatuba was the only municipality which reported two deaths, both in children, in the same year during the study period, 2018. Meanwhile, incidence rates were lowest in the southern, eastern, and coastal regions of SP, as well as in the metropolitan region surrounding the capital city of SP.

Over time, an increase in scorpion envenomation incidence rates can be seen in almost all of the municipalities (S1 Fig) and RHDs of SP (Fig 4), corresponding to the temporal increase identified throughout the study period. This increase was much more pronounced in the western, northwestern, and northern regions of the state. The central region of the state, particularly the Piracicaba RHD (which had more moderate growth), acts as a division between the large increase in envenomations in the west and a smaller increase in the east. Spatially, the deaths were spread out randomly, but for each calendar year, they primarily remained in the central, northern, and western regions of the state (Fig 3).

The Gi* statistic clusters were identified by applying FDR adjustment, so that the agglomerates would be more representative and accurate. This adjustment made it possible to identify whether an area had a higher or lower potential for scorpion stings. Fig 5 depicts two hotspots in red: one in the Barretos and São José do Rio Preto RHDs, and the other in the Araçatuba and Presidente Prudente RHDs, both in 2008. Over time, these high-risk agglomerations grew to include more municipalities, until a single cluster involving these four health districts was formed in 2018. This unification is in line with the higher numbers of scorpion envenomations in the western, northwestern, and northern regions of SP, as well as the temporal increase in incidence rates. Meanwhile, cold spots, or those with lower risks, occurred in the same areas, which were identified as having lower incidence rates, namely the southern, eastern, and coastal regions of the state, as well as the São Paulo metropolitan region.

## Discussion

Reckziegel & Pinto (2014) [17] have already demonstrated the increase in scorpion envenomation, and the subsequent increase in scorpion-related deaths, in Brazil, in their assessment of

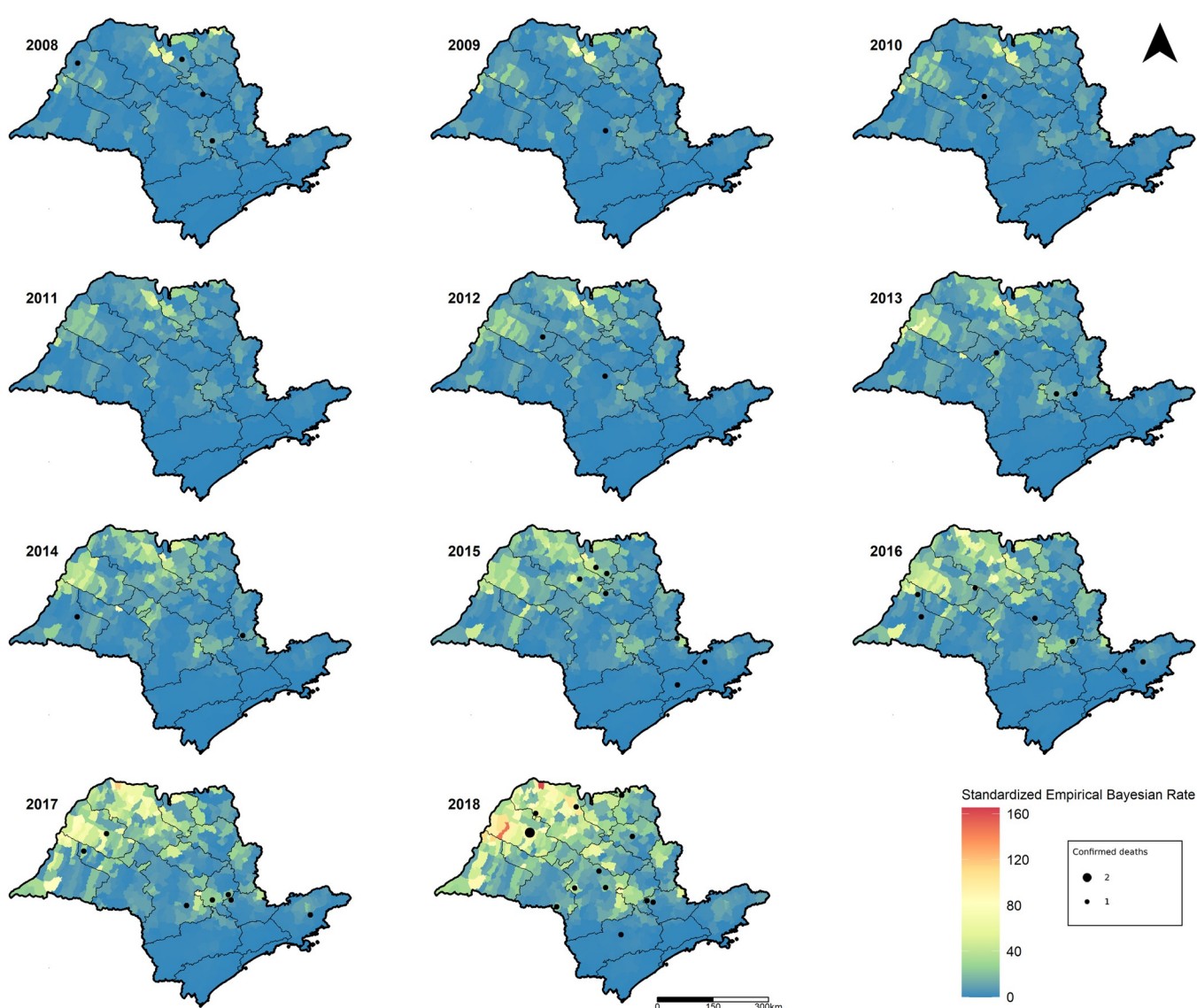

**Fig 3. Maps of standardized local empirical Bayesian incidence rates (per 10,000 inhabitant-years) and deaths from scorpion envenomations for municipalities in the state of São Paulo, Brazil from 2008–2018.**

incidents between 2000 and 2012. They found growth throughout all Brazilian states similar to we found in SP. Additionally, our findings regarding the age group in which envenomation-related deaths occurred are corroborated by other authors in Brazil [17, 32–34], as well relevant international literature [35, 36]. This trend indicates the need for more adequate health measures to reduce the number of scorpion envenomations occurring in these regions, in an effort to avoid deaths in the most vulnerable group (children aged 0–9 years).

The association between the severity of the envenomation and the age group was attributed to the proportion of venom injected in relation to body surface, and a positive association was found between the severity of the envenomation and the amount of venom in the plasma [37]. Reduced levels of circulating vasoactive mediators, such as epinephrine and angiotensin-converting enzyme, were found in children who had experienced a severe envenomation [38]. Additionally, it is possible that the heart and other organs may absorb more venom in this age

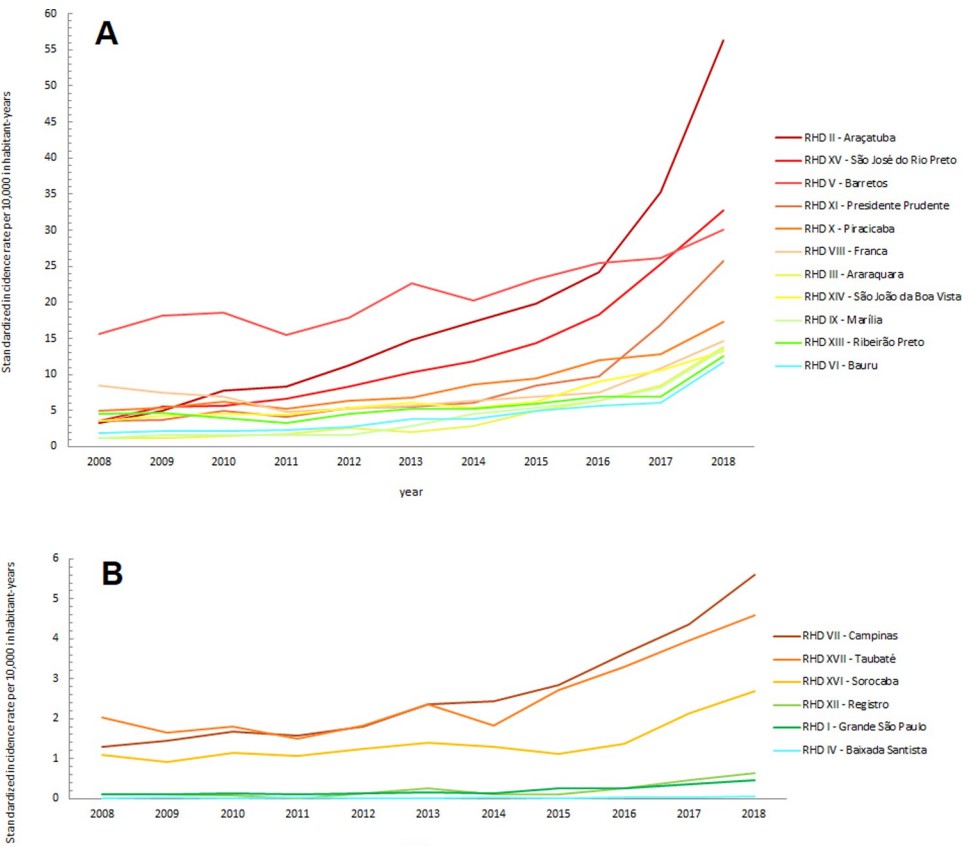

**Fig 4.** Standardized incidence rates of scorpion envenomations (per 10,000 inhabitant-years) for the Regional Health Districts (RHDs) located in the western, northern, northwestern, and central portion (A), and in the eastern and southern portion (B) of the state of São Paulo, from 2008 to 2018.

group [39]. Meanwhile, the occurrence of severe cases among the elderly is most likely due to comorbidities, which affect the cardiovascular system in this population. Regardless, in any age group, a poor prognosis is clearly associated with delays in determining severity and in specific treatment with scorpion or arachnid antivenom.

One reason for the increase in the incidence rate of scorpion envenomations in SP is the continued improvement of the notification system for these accidents, as well as the increased demand for health care from patients affected by scorpion stings in Brazil [40]. The introduction of an internet version of SINAN in 2007 (SINAN Net), for example, improved the operationalization of notifications and the precision of data collection. With the increase in the number of notifications of the accidents and the related deaths, the health system itself became more sensitive, communicating more effectively with the population about the severity of scorpion stings. The media is also involved in this communication circuit, helping to make the population aware of the need for prevention of scorpion accidents, while increasing their sensitivity to the problem, driven them to seek care within the healthcare system when they experience a scorpion sting [40, 41]. The implementation of the scorpion control program in 2009 has played an important role in this scenario [42], as this program has driven improved communication with society, training human resources and the control and environmental management of scorpion infestation throughout Brazilian municipalities (MS 2009).

Additionally, ecological, climate and socioeconomic factors can be associated with the worsening epidemiological picture of scorpion envenomations in SP. A major factor

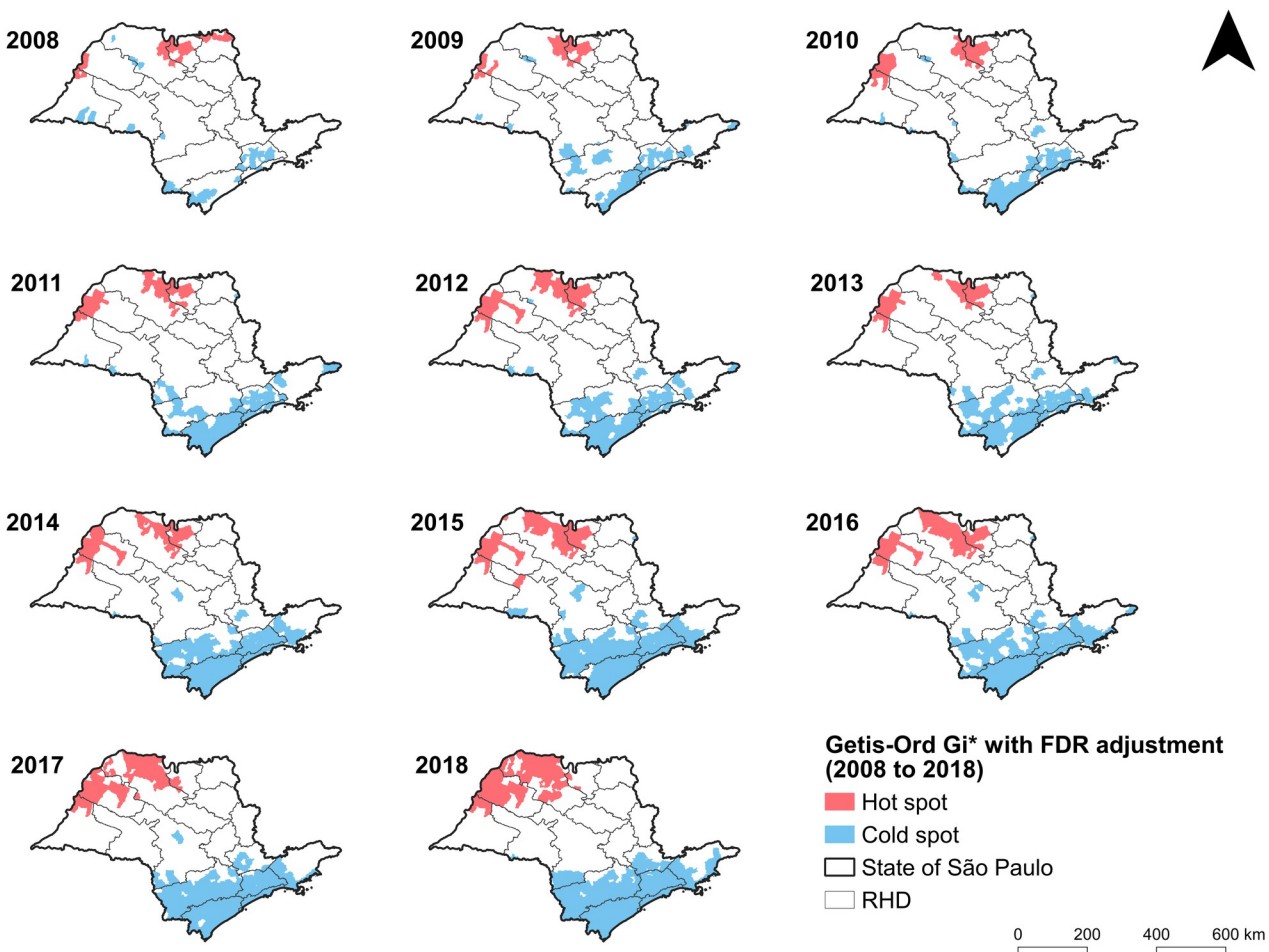

**Fig 5. Spatial clusters of the Getis-Ord Gi* statistic with false discovery rate (FDR) adjustment for municipalities in the state of São Paulo, Brazil from 2008–2018.**

associated with the growing incidence rate of envenomation in SP that we identified in the present study is the replacement of *T. bahiensis* by *T. serrulatus* as the primary scorpion species across nearly the entire state of SP. In a study on the occurrence of medically relevant scorpions in SP, Candido (2008) [43] identified this change, and attributed it primarily to partheno-genesis in *T. serrulatus*. The increase in stings and subsequent deaths may be related to the fact that this species causes more severe envenomations than *T. bahiensis*, and also to a continued increase in *T. serrulatus* infestation [44], which in turn may be related to factors such as urban-ization and climate change. *T. serrulatus* typically colonizes in urban areas (cities and villages), and can be easily unknowingly transported between cities by humans. For example, Brasília was invaded and colonized by *T. serrulatus* in less than 15–20 years [45, 46].

The growing process of urbanization, precarious living conditions, human behavior, and high ecological plasticity of some clinically relevant scorpion species, such as *T. serrulatus* and *T. stigmurus*, may affect the epidemiology of envenomations [47]. Disorderly urban growth can therefore be considered a determining factor in the proliferation of scorpions, where the generation and accumulation of debris plays a fundamental role in the availability of breeding habitats, as these conditions facilitate the spread of cockroaches and other insects that are pri-mary food sources for scorpions [48]. Expanding agricultural frontiers combined with clearing

of native vegetation also contributes to the higher incidence of scorpion stings, since the destruction of scorpions' natural habitats and that of their natural predators (such as monkeys, raccoons, frogs, owls, and lizards) causes an ecological imbalance followed, by a subsequent boom in scorpion populations [17, 49].

The positive selection of harmful opportunistic species is directly associated with human activity. In profoundly modified environments, such as several "artificial" cities in Brazil, the human population begins to grow rapidly, and the three main factors required for high incidences of scorpion envenomations in these regions became present: first, the demographic expansion of the human population; second, the rapid expansion of populations of harmful opportunistic scorpions that can occupy the empty niches left behind when species in equilibrium diminish or go extinct, and in many cases, opportunistic species adapt their behavior and move into human dwellings; and third, the overlap between a large human population and a large population of harmful scorpions greatly increases the likelihood of scorpion stings [50], a situation which is typical in several regions of Brazil, although primarily in the southeast and Midwest regions.

An additional factor that may be related to the increase in scorpions, and therefore scorpion stings, is climate change, which affects the entire planet. It has severely affected the environment, and may influence the distribution of arthropod populations. In a recent study on how climate affects scorpions in Iran, Rafinejad et al. (2020) [25] emphasized that climate change is an important variable in the spatial distribution of these arthropods, because their activities are highly dependent on environmental conditions. For example, higher temperatures may shorten the procreation times while increasing the maturation rate of scorpions.

Needleman et al. (2018) [51] stated that climatic effects are closely linked to land-based venomous species, and that environmental changes could result in greater species migration, geographical redistribution, and longer periods with more stings, which would have repercussions for human health. All three of these hypothetical impacts have been seen in SP. The first was the near-total replacement of *T. bahiensis* (which had been the predominant species) by *T. serrulatus*, which is parthenogenetic and successfully adapted to profoundly modified environments [43, 52]. Braga-Pereira & Santos (2021) [53] also showed that parthenogenetic reproduction in *T. serrulatus* can occur not only in asexual populations of this species, but also in those where sexual reproduction occurs—in other words, parthenogenesis is optional in *T. serrulatus*. The second and third hypotheses raised by Needleman et al. (2018) [51], regarding geographical redistribution and a continued increase in the number of scorpion stings, also correspond to the results of the present study. Abreu et al. (2019) [54] noted that higher temperatures in southeast Brazil between 1995 and 2004 would have been appropriate for the proliferation of scorpions in this region. The findings of the present study confirm the increased incidence of scorpion stings in almost all municipalities and regions of SP, while also demonstrating that these incidences are not evenly distributed in the municipalities in which they occur. At they are more numerous in the western, northwestern, and northern regions. These regions are part of the western plateau of SP, and have the highest temperatures and lowest rainfall in SP [27], which favors the development of scorpions [25, 55]. Ureta et al. (2020) [56] also studied the impacts of climate change on scorpions, using ecological niche modeling in Mexico, from whichthey identified the dispersion and geographical redistribution of some scorpion species among the country's regions.

In an analysis of notifications and government outreach services resulting from scorpion sightings by the public in SP, Morais et al. (2021) [44] found higher numbers of scorpions in the western, northwestern, and northern regions of the state. The monitoring system revealed higher numbers of scorpion notifications in the Ribeirao Preto, Araçatuba, Presidente Prudente, Barretos, and São José do Rio Preto RHDs, and lower numbers in the Baixada Santista

RHD. In a study of scorpion stings in SP between 2000 and 2011 with future predictions, Aze-vedo et al. (2019) [49], also noted an increased probability of envenomations due to climate change, and authors showed that this increase is related to changes in temperature amplitude, and is therefore expected to be more pronounced in the western, northwestern, and northern regions of the state; specifically the regions we found to be most affected.

There are, of course, limitations to the present study. The first is that it is based on second-ary data on passive notifications of scorpion stings. Two possible biases are relate to this point: the notification system is subject to underreporting, which can generate incidences lower than the actual values, although the system has improved over time, with is reflected both in the increase in notifications and in the demand for health care [40]. These are important limita-tions, as they were not considered in the present study.

Second, we did not consider data regarding the notification of scorpion species in SP. These data have recently been registered in a system created by the Endemic Diseases Department (SUCEN), a section of the São Paulo State Department of Health, to monitor these arthropods in the state. This system identifies both notifications and captures of scorpions by municipal technicians in real-time, and provides the geographical distribution of the species present throughout the state, generating indicators that are useful for the planning of municipal con-trol and management activities related to scorpions [44].

There are several factors involved in epidemiology of scorpionism that could be futher explored to better understand this health problem: information about the scorpion population, environmental, demographic and socioeconomic conditions; climate change; accessibility to health care; and differences in the scorpion control methods and the management of scorpion stings by the municipalities; among others. Even though we previously discussed possible fac-tors associated with scorpion envenomation, it is a limitation of the present study the fact that we do not consider these factors in our analysis.

The primary strength of our study is the use of spatial analysis techniques to map scorpion stings and detect at-risk areas. For mapping, we utilized the local empirical Bayesian rate to control random fluctuations of data in areas with small populations. To detect clusters, $Gi^*$ [57] with FDR correction was used for multiple comparisons. This technique, which limits the possibility of finding false-positive results, is more precise in identifying real agglomerates [31, 58]. Another strength of the present study is the standardization of the rates according to age and sex. This is an important factor for avoiding confusion bias in studies due to population age differences. Furthermore, this technique is often not described in the literature when the local empirical Bayesian method is used [59].

The best of our knowledge, this is the first study to highlight the increasing magnitude of scorpionism in the state of SP. Future studies should investigate the environmental and cli-matic characteristics of the areas with high and low rates of scorpion stings, and model the incidence rates of scorpionism and scorpion occurrence using environmental, climatic, demo-graphic, socioeconomic, and other municipatity-specific characteristics. Therefore, it is crucial to improve the surveillance and control of this growing health concern in SP. Although we did not consider these characteristics in the present study, it is an important first step in under-standing the actual distribution of scorpionism in SP, and can serve as a basis for future studies.

## Conclusions

The incidence and mortality rates of scorpion envenomations increased significantly from 2008 to 2018 in the state of SP. Although this increase in incidence occurred in almost all municipalities and regions of the state, the highest incidences were found in the western,

northwestern, and northern regions. Incidence rates were higher for men, and increased with age; however, fatalities from envenomations were concentrated almost entirely in children 0–9 years old of age. The distribution of accidents found in SP, as well as the identification of areas of greater risk, can be used to identify priority areas for the development of surveillance and control plans, both at the regional and municipal level. This distribution could also be useful in assessing the suitability of the location of the reference units for the care of scorpion accidents in SP, and may also be useful for sizing antivenom serum needs. The findings of the present study can help health services make decisions to reduce contact with scorpions and avoid fatalities, especially among children.

## Supporting information

**S1 File. Scorpionism data aggregated by municipality, year, sex and age group with local empirical Bayesian standardization and incidence rates.**
(XLSX)

**S1 Fig. Boxplot of local standardized empirical Bayesian rates (per 10,000 inhabitant-years) by year for all municipalities of the state of São Paulo, Brazil.**
(TIF)

## Author Contributions

**Conceptualization:** Alec Brian Lacerda, Thiago Salomão De Azevedo, Denise Maria Cândido, Fan Hui Wen, Luciano José Eloy, Ana Aparecida Sanches Bersusa, Francisco Chiaravalloti Neto.

**Data curation:** Alec Brian Lacerda, Camila Lorenz, Thiago Salomão De Azevedo, Luciano José Eloy.

**Formal analysis:** Alec Brian Lacerda.

**Funding acquisition:** Francisco Chiaravalloti Neto.

**Methodology:** Camila Lorenz, Thiago Salomão De Azevedo, Denise Maria Cândido, Fan Hui Wen, Luciano José Eloy, Ana Aparecida Sanches Bersusa, Francisco Chiaravalloti Neto.

**Project administration:** Francisco Chiaravalloti Neto.

**Supervision:** Francisco Chiaravalloti Neto.

**Validation:** Alec Brian Lacerda, Camila Lorenz.

**Visualization:** Alec Brian Lacerda.

**Writing – original draft:** Alec Brian Lacerda, Camila Lorenz, Thiago Salomão De Azevedo, Denise Maria Cândido, Fan Hui Wen, Luciano José Eloy, Ana Aparecida Sanches Bersusa, Francisco Chiaravalloti Neto.

**Writing – review & editing:** Alec Brian Lacerda, Camila Lorenz, Thiago Salomão De Azevedo, Denise Maria Cândido, Fan Hui Wen, Luciano José Eloy, Ana Aparecida Sanches Bersusa, Francisco Chiaravalloti Neto.

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
