## [Decision Letter · Decision Letter 0]

26 Dec 2021

PONE-D-21-37124Scorpion envenomation in the state of São Paulo, Brazil: Spatiotemporal analysis of a growing public health problemPLOS ONE

Dear Dr. Lacerda,

Thank you for submitting your manuscript to PLOS ONE. After careful consideration, we feel that it has merit but does not fully meet PLOS ONE’s publication criteria as it currently stands. Therefore, we invite you to submit a revised version of the manuscript that addresses the points raised during the review process.

ACADEMIC EDITOR: Reviewers valued this contribution as it describes a relevant and interesting public health phenomenon, i.e., the increment in the incidence of scorpion stings and their derived envenomings in the state of Sao Paulo, Brazil. Reviewers also highlighted a number of issues that need to be taken into consideration when preparing a revised version of this manuscript. The most important aspects to consider are: (a) the need to make the data of the study publicly available, as per the policy of the journal, (b) the need to discuss the limitations of the methodology used andthe conclusions reached, and (c) the strengthening of the discussion by considering some aspects mentioned by the reviewers, such as the possible geographical differences within the state regarding incidence and mortality due to scorpion sting envenomings, and further exploring and discussing the hypothesis presented to explain the increment in the incidence of these accidents, among other points.  

We look forward to receiving your revised manuscript.

Kind regards,

José María Gutiérrez

Academic Editor

PLOS ONE

Journal Requirements:

Additional Editor Comments:

3. We note that Figures 1, 3 and 4 in your submission contain [map/satellite] images which may be copyrighted. All PLOS content is published under the Creative Commons Attribution License (CC BY 4.0), which means that the manuscript, images, and Supporting Information files will be freely available online, and any third party is permitted to access, download, copy, distribute, and use these materials in any way, even commercially, with proper attribution. For these reasons, we cannot publish previously copyrighted maps or satellite images created using proprietary data, such as Google software (Google Maps, Street View, and Earth). For more information, see our copyright guidelines: http://journals.plos.org/plosone/s/licenses-and-copyright.

a) You may seek permission from the original copyright holder of Figures 1, 3 and 4 to publish the content specifically under the CC BY 4.0 license.  

4.  The reviewers appreciated the relevance of this study, since it focuses on an important public health issue in the state of Sao Paulo, i.e., the drastic increment in the incidence of envenomings by scorpion stings, a phenomenon that deserves analysis and attention by the research community and the public health system of the state. Reviewers highlighted a number of issues that should be carefully considered and taken into consideration for preparing a revised version of the manuscript.

Reviewers' comments:

Reviewer's Responses to Questions

**Comments to the Author**

1. Is the manuscript technically sound, and do the data support the conclusions?

Reviewer #1: Yes

Reviewer #2: Partly

Reviewer #3: Yes

2. Has the statistical analysis been performed appropriately and rigorously? 

Reviewer #1: Yes

Reviewer #2: I Don't Know

Reviewer #3: Yes

3. Have the authors made all data underlying the findings in their manuscript fully available?

Reviewer #1: No

Reviewer #2: No

Reviewer #3: Yes

4. Is the manuscript presented in an intelligible fashion and written in standard English?

Reviewer #1: No

Reviewer #2: Yes

Reviewer #3: Yes

5. Review Comments to the Author

Reviewer #1: This study analyzed the historical series of scorpion stings in São Paulo, Brazil from 2008 to 2018 and occurrences in the 645 municipalities of this state.

Abstract:

1. "standardized by sex and age": change to "stratified by sex and age"

2. "the deaths were randomly distributed": randomly or proportional to incidence?

3. "Our findings that identify areas and populations at risk for scorpion envenomation and resulting fatalities can...": Change sentence. Suggestion: "In this study, we identified areas and population at risk for SE and associated-fatalities, which can..."

4. "...with these animals and avoid fatalities, especially in the most vulnerable population." Why mostly in the vulnerable population? Is it possible to conclude that from the results?

Introduction:

1. "Tityus is most medically relevant due to envenomations in children and high incidences in recent years." The other genera do not sting children? Strange sentence.

2. "Four species are epidemiologically important in Brazil: Tityus serrulatus, T. bahiensis, T. obscurus, and T. stigmurus." Sentence needs reference.

3. "Antivenin": Change to "antivenom" in the whole text.

4. "Antivenin is recommended for moderate and severe cases. [10, 13, 14]". Where? In Brazil? Accornding official guidelines?

5. "serious cases": change to "severe cases" in the whole text.

6. "at least minimize the sequelae and/or deaths". Sequelae? Which types?

7. "For this reason, it is extremely important 113 to characterize scorpion exposures in terms of victims as well as spatiotemporal factors." What 'in terms of victims' mean?

8. "This study consequently describes": Delete 'consequently'.

Methods

1. "scorpion exposures": change to "scorpion stings"in the whole text.

2. Include environmental and climate information in the Study area section.

3. "institutional ethics review board of the University of São Paulo School of Public Health (COEP FSP/USP, CAAE approval record 10457119.6.0000.5421, opinion 3408558)". Opinion?

Discussion:

"These authors showed that this increase is related to changes in temperature amplitude and is expected to be more pronounced in the north and northwest of the state, precisely the regions we found to be most affected."

I believe that the discussion about the greater incidence in the northern and northwestern regions of the state counties deserves further attention. What characteristics do these regions have that differentiate them from the rest of the state, in climatic and environmental terms, but also in terms of economic activities that they could include in contact with the scorpions? Different biomas? Are there studies on the density of these arthropods in the state? What do these works show for the regions with higher incidence? Different human development indexes?

Reviewer #2: The manuscript “Scorpion envenomation in the state of São Paulo, Brazil: Spatiotemporal analysis of a growing public health problem” by Lacerda et al. shows a dramatic increase in the incidence and mortality from scorpion envenomations between 2008 and 2018. The study aims to identify areas and populations at risk of scorpion envenomation and death in order to promote decision-making by the health services.

My main concern is that the authors do not sufficiently discuss the limitations of their study, especially with regard to the collection and analysis of data provided by the São Paulo State Epidemiological Surveillance Center.

The methodology is relevant and well described.

The results are aggregated, which does not allow a precise view of the data by group of people (sex, age), region and their evolution over time. Data confidentiality must be balanced for with more detailed analysis and comparisons.

The discussion raises 3 hypotheses – classic but convincing – about the causes of the increase in incidence and mortality, in particular the expansion of T. serrulatus which is gradually replacing T. bahiensis due to the new environmental conditions, and the evolution of human activities leading to both attraction of scorpions and increasing exposure. In this regard, the evolution of human activity is likely to be more strongly involved than climate change, the impact of which would be more gradual and could not explain, for example, the sudden increase in incidence and mortality from 2014.

However, as I indicated above, the authors do not sufficiently address other possible explanations such as the insufficient reporting of cases which would have improved between 2008 and 2018 or a gradual evolution of the healthcare seeking behavior of the patients (see Chippaux. J Venom Anim Toxins Incl Trop Dis. 2015; 21:13). They do not seek to verify these quite surprising results using data provided by independent sources.

Two ways should be considered to validate these hypotheses.

On the one hand, the authors indicate that the risk increased in all regions of the state of São Paulo throughout the study period. First, it is necessary to verify that the variables of interest show similar or even parallel evolutions in the different parts of the state and that there are no areas of significantly higher incidence and / or mortality which could be explained by variations in environmental conditions (scorpion population, human activities) and / or the management of scorpion stings (specially to prevent severe outcomes or death). There can be no convergence of epidemiological parameters (in particular parallelism of incidence in different places) in all the regions resulting from the authors' hypothesis insofar as the situations and their consequences cannot be identical in all the regions of the state of São Paulo. However, the authors indicate that the increase is higher (without data being available) in some regions. They should compare the explanatory situations (abundance of T. serrulatus, human activities, environmental and sanitary conditions, accessibility of health facilities, climate change) between different regions showing the highest differences in incidence (e.g. between the northwest, center and southern regions of the state of São Paulo).

On the other hand, the authors should check whether the increase in incidence and case fatality rate observed during the same period in randomly selected health centers in different regions is similar to that reported by the São Paulo State Epidemiological Surveillance Center.

I disagree with the authors when they state that underreporting of scorpion stings, especially mild cases, can be considered to occur uniformly in the state of São Paulo. The reasons for underreporting are not equivalent depending on the place and the person in charge of the case report. The case report may be affected by incidence, particularly frequency of mild or moderate envenomation, perception of the case severity, individual interest for the topic, overwork, etc...

Moreover, the authors do not seek to explain the increase in mortality, nor the random distribution of deaths between regions. They should study the case fatality rate of scorpion envenomation in a few hospitals located in areas with higher mortality. The increased case fatality rate could be related to difficulties in managing envenomation, e.g. failure of the antivenom supply or lack of knowledge of treatment of scorpion envenomation.

Assuming that the authors cannot answer some of these questions, they could at least mention and argue them.

Finally, the authors do not offer practical recommendations and let the health authorities make any decision based on their results. I think it's up to the authors to guide health authorities based on their results and knowledge of the problem.

Reviewer #3: The article addresses a very important issue related to scorpion injuries in Brazil, and also in other regions of the world, which has been increasing progressively year by year. The object of the study was the state of São Paulo, located in southeastern Brazil and which has 645 municipalities divided into 17 regional health districts, from 2008 to 2018. During this period there was an increase in the number of accidents of 425%. The aim of the study was to analyze the temporal and spatial distribution of the occurrence of accidents to identify areas of greater or lesser risk in order to develop strategies to reduce the population's contact with these scorpions to avoid deaths in the most vulnerable group (children 0-9 years of age).

The authors calculated the annual incidence and mortality rate standardized by sex and age. They used the empirical local Bayesian method and Gi* statistics to standardize the incidence rates in the municipalities and identify high and low risks of agglomeration.

There was a higher incidence and mortality among men, but mortality was concentrated in children under 9 years of age, with 42 of the total 48 deaths (88%) occurring in this group. From the analysis obtained by the maps, it was observed that accidents occurred more in the northern and northwestern regions of the state in contrast to the other regions, in addition to showing that the incidence of cases increased in all regions of S.Paulo.

The discussion about why this is happening was quite comprehensive, in terms of increased urbanization, climate effects, housing conditions, poverty, T. serrulatus parthenogenesis, etc.

6. PLOS authors have the option to publish the peer review history of their article (what does this mean?). If published, this will include your full peer review and any attached files.

Reviewer #1: No

Reviewer #2: **Yes: **Jean-Philippe Chippaux

Reviewer #3: **Yes: **Palmira Cupo

---

## [Author Response · Author response to Decision Letter 0]

3 Mar 2022

Dear editor,

We are submitting a second version of the manuscript entitled "Scorpion envenomation in the state of São Paulo, Brazil: Spatiotemporal analysis of a growing public health concern". We sent our manuscript to Editage for an English review and we accepted their suggestion to change our title (the old one was “Scorpion envenomation in the state of São Paulo, Brazil: Spatiotemporal analysis of a growing public health problem”). We have made substantial changes in the manuscript and have addressed the points raised by the reviewers. The changes are summarized below (in bold) to each reviewer's comment. We indicated the position (lines) of our modifications using the 'Revised Manuscript with Track Changes' version of our manuscript. We would like to thank the reviewers for their helpful comments and suggestions. We further thank you for your valuable time spared for our manuscript.

A - ANSWERS TO EDITOR

1 - The most important aspects to consider are: 

(a) the need to make the data of the study publicly available, as per the policy of the journal, 

Answer: We included the database as Supplementary Material 1 (S1 File).

(b) the need to discuss the limitations of the methodology used and the conclusions reached, and 

Answer: We improve the discussion of the limitations of our study (lines 430-460 of Track Changes version; lines 388-410 of Manuscript version).

(c) the strengthening of the discussion by considering some aspects mentioned by the reviewers, such as the possible geographical differences within the state regarding incidence and mortality due to scorpion sting envenomings, and further exploring and discussing the hypothesis presented to explain the increment in the incidence of these accidents, among other points. 

Answer: We improved our Discussion, taking into account these issues.

2 – Figures of our manuscript

Answer: We created the Figures 1, 3 and 5 (old Figure 4) using data and shapefiles from the Brazilian Institute of Geography and Statistics (IBGE) and the Brazilian Information System for Notifiable Diseases (SINAN). This information is of free and open accesses for everyone. So the maps we use in our manuscript, since they are freely accessible, are not copyrighted. Moreover, these maps were created using the Free and Open Source QGIS.

B - ANSWERS TO REVIEWERS

 Reviewers' comments:

III.1 - Reviewer's Responses to Questions

Comments to the Author

1. Is the manuscript technically sound, and do the data support the conclusions?

Reviewer #1: Yes

Reviewer #2: Partly

Reviewer #3: Yes

2. Has the statistical analysis been performed appropriately and rigorously?

Reviewer #1: Yes

Reviewer #2: I Don't Know

Reviewer #3: Yes

3. Have the authors made all data underlying the findings in their manuscript fully available?

Reviewer #1: No

Reviewer #2: No

Reviewer #3: Yes

Answer: We made our data available in the Supplementary Material 1 (S1 File).

4. Is the manuscript presented in an intelligible fashion and written in standard English?

Reviewer #1: No

Reviewer #2: Yes

Reviewer #3: Yes

Answer: We made the English correction suggested by the Reviewers and also sent our manuscript to Editage for an English review. We attached the certificate of this review in the submission system.

III.2 - Review Comments to the Author

A - Reviewer #1: This study analyzed the historical series of scorpion stings in São Paulo, Brazil from 2008 to 2018 and occurrences in the 645 municipalities of this state.

Abstract:

1. "standardized by sex and age": change to "stratified by sex and age"

Answer: Ok, we corrected this (line 35).

2. "the deaths were randomly distributed": randomly or proportional to incidence?

Answer: According to our results, we did not see a directly correspondence between the deaths and the incidence rates, since the deaths occurred in all areas (high, median and low incidence rates). Then, we maintained the text in its original presentation.

3. "Our findings that identify areas and populations at risk for scorpion envenomation and resulting fatalities can...": Change sentence. Suggestion: "In this study, we identified areas and population at risk for SE and associated-fatalities, which can..."

Answer: We accepted this suggestion and changed our text (lines 49-53 of Track Changes version; lines 47-50 of Manuscript version).

4. "...with these animals and avoid fatalities, especially in the most vulnerable population." Why mostly in the vulnerable population? Is it possible to conclude that from the results?

Answer: We found that the most vulnerable population are the 0 to 9 years old children, with the highest mortality rates. We changed the text to “…with these arachnids and avoid fatalities, especially in children.” (lines 52-53 of Track Changes version; lines 49-50 of Manuscript version).

Introduction:

1. "Tityus is most medically relevant due to envenomations in children and high incidences in recent years." The other genera do not sting children? Strange sentence.

Answer: We changed the text to “Tityus is most medically relevant scorpion genus due to clinical manifestations caused by envenomations in human and the high incidence in recent years." (lines 70-72 of Track Changes version; lines 67-69 of Manuscript version). 

2. "Four species are epidemiologically important in Brazil: Tityus serrulatus, T. bahiensis, T. obscurus, and T. stigmurus." Sentence needs reference. 

Answer: We inserted two references about this issue (line 74 of Track Changes version; line 71 of Manuscript version).

3. "Antivenin": Change to "antivenom" in the whole text.

Answer: We changed the term “antivenin” to “antivenom” in the whole text.

4. "Antivenin is recommended for moderate and severe cases. [10, 13, 14]". Where? In Brazil? Accornding official guidelines?

Answer: We include the following phrase to explain better this issue: “According to the Brazilian Ministry of Health Guidelines, antivenom treatment is recommended in which the patient presents with signs and symptoms of systemic envenomation, which are classified as moderate or severe cases depending on the severity of the clinical manifestations.” (lines 87-91 of Track Changes version; lines 81-85 of Manuscript version). 

5. "serious cases": change to "severe cases" in the whole text.

Answer: We changed the term “serious” to “severe” in the whole text.

6. "at least minimize the sequelae and/or deaths". Sequelae? Which types?

Answer: We changed the text to “at least minimize the risk of death due to an envenomation.” (lines 105-106 of Track Changes version; lines 97-98 of Manuscript version).

7. "For this reason, it is extremely important to characterize scorpion exposures in terms of victims as well as spatiotemporal factors." What 'in terms of victims' mean?

Answer: We changed the text to "For this reason, it is extremely important to characterize scorpion envenomations in terms of the people affected by scorpion stings, as well as, spatiotemporal factors." (lines 106-109 of Track Changes version; lines 98-100 of Manuscript version).

8. "This study consequently describes": Delete 'consequently'.

Answer: Ok we corrected this (line 115 of Track Changes version; line 105 of Manuscript version).

Methods

1. "scorpion exposures": change to "scorpion stings" in the whole text.

Answer: We did this in the whole text.

2. Include environmental and climate information in the Study area section.

Answer: We included a new paragraph (2nd paragraph of the Methods) to deal with this issue. We changed the Figure 1, including the relief of the state of São Paulo to better explain its climate information (lines 131-145 of Track Changes version; lines 120-134 of Manuscript version).

3. "institutional ethics review board of the University of São Paulo School of Public Health (COEP FSP/USP, CAAE approval record 10457119.6.0000.5421, opinion 3408558)". Opinion?

Answer: We change ‘opinion’ to ‘protocol number’ (line 193 of Track Changes version; line 176 of Manuscript version).

Discussion:

"These authors showed that this increase is related to changes in temperature amplitude and is expected to be more pronounced in the north and northwest of the state, precisely the regions we found to be most affected."

I believe that the discussion about the greater incidence in the northern and northwestern regions of the state counties deserves further attention. What characteristics do these regions have that differentiate them from the rest of the state, in climatic and environmental terms, but also in terms of economic activities that they could include in contact with the scorpions? Different biomas? Are there studies on the density of these arthropods in the state? What do these works show for the regions with higher incidence? Different human development indexes?

Answer: We formed a team to study the scorpions and their accidents in the state of São Paulo and to do propositions to improve the surveillance and control of this growing health problem in our state. The team includes technicians and researchers of the São Paulo State Epidemiological Surveillance Center (CVE), Butantan Institute, and Endemic Diseases Department (SUCEN), which are sections of the São Paulo State Department of Health. This team also includes people of the School of Public Health of the University of São Paulo and of the Department of Health of the Municipality of Santa Bárbara d’Oeste, state of São Paulo. This manuscript is one of the first to be produced and we are conducting other studies where we are investigating the environmental and climatic characteristics of the cluster of high and low rates of scorpion stings in of the state of São Paulo, and modelling the incidence rates of scorpionism and the scorpion occurrence using environmental, climatic, demographic, socioeconomic and other characteristics of the municipalities of our state. Since we had a broad list of goals in the present manuscript and we are preparing other manuscripts, we considered that it would not be productive or viable to include results related to covariates associated to the scorpionism. On the other hand, we did not fail to present, in our discussion section, the factors possibly associated with the phenomenon studied. We changed our Discussion to deal better with these issues. We also included a new paragraph at the end of the Discussion about future studies and the need to investigate the factors associated with the scorpion stings and the scorpion infestation (lines 471-479 of Track Changes version; lines 421-429 of Manuscript version).

B - Reviewer #2 – Dr. Jean-Philippe Chippaux: The manuscript “Scorpion envenomation in the state of São Paulo, Brazil: Spatiotemporal analysis of a growing public health problem” by Lacerda et al. shows a dramatic increase in the incidence and mortality from scorpion envenomations between 2008 and 2018. The study aims to identify areas and populations at risk of scorpion envenomation and death in order to promote decision-making by the health services.

1 - My main concern is that the authors do not sufficiently discuss the limitations of their study, especially with regard to the collection and analysis of data provided by the São Paulo State Epidemiological Surveillance Center.

Answer: We rewrote and expanded the text about the limitations of our study (lines 430-460 of Track Changes version; lines 388-410 of Manuscript version).

2 - The methodology is relevant and well described.

Answer: Thank you for recognizing the relevance of the Methods of our manuscript.

3 - The results are aggregated, which does not allow a precise view of the data by group of people (sex, age), region and their evolution over time. Data confidentiality must be balanced for with more detailed analysis and comparisons.

Answer: We included a new figure (Figure 4 – its title is on lines 264-267 of Track Changes version; lines 242-245 of Manuscript version) that presents the evolution of the scorpion stings incidence rates from 2008 to 2018 for the 17 Regional Health Districts (RHDs) of the state of São Paulo. In addition to detailing the presentation of the incidence rates of scorpion envenomations according to regions of the state, another motivation is to show that the increase of the incidence rates occurred in almost all regions. This also happened with the municipalities, as it is possible to see in the Supplementary Material 2 that we included in this new version. Table 1 presents the scorpion envenomation for groups of people by sex and age. Figure 1 presents the temporal evolution of the incidence and mortality rates of scorpion stings for the state of São Paulo during 2008 to 2018.

4 – The discussion raises 3 hypotheses – classic but convincing – about the causes of the increase in incidence and mortality, in particular the expansion of T. serrulatus which is gradually replacing T. bahiensis due to the new environmental conditions, and the evolution of human activities leading to both attraction of scorpions and increasing exposure. 

4.1 - In this regard, the evolution of human activity is likely to be more strongly involved than climate change, the impact of which would be more gradual and could not explain, for example, the sudden increase in incidence and mortality from 2014. However, as I indicated above, the authors do not sufficiently address other possible explanations such as the insufficient reporting of cases which would have improved between 2008 and 2018 or a gradual evolution of the healthcare seeking behavior of the patients (see Chippaux. J Venom Anim Toxins Incl Trop Dis. 2015; 21:13). They do not seek to verify these quite surprising results using data provided by independent sources.

Answer: We agree with this observation and included a new paragraph in the Discussion to deal with these issues (3rd paragraph of Discussion section – lines 325-339 of Track Changes version; lines 294- 308 of Manuscript version). Thank you for the indication of this paper.

4.2 - Two ways should be considered to validate these hypotheses.

On the one hand, the authors indicate that the risk increased in all regions of the state of São Paulo throughout the study period. First, it is necessary to verify that the variables of interest show similar or even parallel evolutions in the different parts of the state and that there are no areas of significantly higher incidence and / or mortality which could be explained by variations in environmental conditions (scorpion population, human activities) and / or the management of scorpion stings (specially to prevent severe outcomes or death). There can be no convergence of epidemiological parameters (in particular parallelism of incidence in different places) in all the regions resulting from the authors' hypothesis insofar as the situations and their consequences cannot be identical in all the regions of the state of São Paulo. However, the authors indicate that the increase is higher (without data being available) in some regions. They should compare the explanatory situations (abundance of T. serrulatus, human activities, environmental and sanitary conditions, accessibility of health facilities, climate change) between different regions showing the highest differences in incidence (e.g. between the northwest, center and southern regions of the state of São Paulo).

On the other hand, the authors should check whether the increase in incidence and case fatality rate observed during the same period in randomly selected health centers in different regions is similar to that reported by the São Paulo State Epidemiological Surveillance Center.

I disagree with the authors when they state that underreporting of scorpion stings, especially mild cases, can be considered to occur uniformly in the state of São Paulo. The reasons for underreporting are not equivalent depending on the place and the person in charge of the case report. The case report may be affected by incidence, particularly frequency of mild or moderate envenomation, perception of the case severity, individual interest for the topic, overwork, etc...

Moreover, the authors do not seek to explain the increase in mortality, nor the random distribution of deaths between regions. They should study the case fatality rate of scorpion envenomation in a few hospitals located in areas with higher mortality. The increased case fatality rate could be related to difficulties in managing envenomation, e.g. failure of the antivenom supply or lack of knowledge of treatment of scorpion envenomation.

Assuming that the authors cannot answer some of these questions, they could at least mention and argue them.

Answer: We included two figures in this new version of our manuscript. Figure 4 (its title is on lines 264-267 of Track Changes version; lines 242-245 of Manuscript version) shows that the increase of the incidence rates of scorpion stings occurred in almost all Regional Health Districts (RHDs) of the state of São Paulo. We also introduced a Figure, as Supplementary Material 2 (S2 Fig), showing that the increase in the incidence rates occurred, similar to what happened with the regions, in almost all municipalities of the state. Even revealing that this increase was a widespread phenomenon in the state, the methods we used (Local empirical Bayesian method and Getis-Ord Gi*) showed that it was more pronounced in the western, northwestern, and northern regions of SP, with significant high risk clusters. 

We agree that there are several factors involved in the scorpionism epidemiology that could be explored to better understand this important public health, as the factors cited in this review. We discussed part of them, but did not take them into account in our analysis. So that, we considered this another limitation of our study (lines 453-460 of Track Changes version; lines 403-410 of Manuscript version). Nonetheless, we are conducting new studies considering these factors in our analysis. As we had pointed out to the Review # 1, we formed a team to study the scorpions and their accidents in SP and to do propositions to improve the surveillance and control of this important health problem in our state. The team includes technicians and researchers of the São Paulo State Epidemiological Surveillance Center (CVE), Butantan Institute, and Endemic Diseases Department (SUCEN), which are sections of the São Paulo State Department of Health. This team also includes technicians and researchers of the School of Public Health of the University of São Paulo and of the Department of Health of the Municipality of Santa Bárbara d’Oeste, state of São Paulo. This manuscript is one of the first to be produced and we are conducting other studies where we are investigating the environmental and climatic characteristics of the cluster of high and low rates of scorpion stings in of the state of São Paulo, and modelling the incidence rates of scorpionism and the scorpion occurrence using environmental, climatic, demographic, socioeconomic and other characteristics of the municipalities of our state.

We agreed with the reviewer about our statement that underreporting of scorpion stings, especially mild cases, can be considered to occur uniformly in the state of São Paulo and remove this text of the Discussion section.

4.4 - Finally, the authors do not offer practical recommendations and let the health authorities make any decision based on their results. I think it's up to the authors to guide health authorities based on their results and knowledge of the problem.

Answer: We included these recommendations in the Conclusions (lines 492-499 of Track Changes version; lines 437-444 of Manuscript version).

 Reviewer #3 – Dr. Palmira Cupo: The article addresses a very important issue related to scorpion injuries in Brazil, and also in other regions of the world, which has been increasing progressively year by year. The object of the study was the state of São Paulo, located in southeastern Brazil and which has 645 municipalities divided into 17 regional health districts, from 2008 to 2018. During this period there was an increase in the number of accidents of 425%. The aim of the study was to analyze the temporal and spatial distribution of the occurrence of accidents to identify areas of greater or lesser risk in order to develop strategies to reduce the population's contact with these scorpions to avoid deaths in the most vulnerable group (children 0-9 years of age).

The authors calculated the annual incidence and mortality rate standardized by sex and age. They used the empirical local Bayesian method and Gi* statistics to standardize the incidence rates in the municipalities and identify high and low risks of agglomeration.

There was a higher incidence and mortality among men, but mortality was concentrated in children under 9 years of age, with 42 of the total 48 deaths (88%) occurring in this group. From the analysis obtained by the maps, it was observed that accidents occurred more in the northern and northwestern regions of the state in contrast to the other regions, in addition to showing that the incidence of cases increased in all regions of S.Paulo.

The discussion about why this is happening was quite comprehensive, in terms of increased urbanization, climate effects, housing conditions, poverty, T. serrulatus parthenogenesis, etc.

Answer: Thank you for your comments and for considering our study suitable and important.

---

## [Decision Letter · Decision Letter 1]

15 Mar 2022

Scorpion envenomation in the state of São Paulo, Brazil: Spatiotemporal analysis of a growing public health concern

PONE-D-21-37124R1

Dear Dr. Lacerda,

We’re pleased to inform you that your manuscript has been judged scientifically suitable for publication and will be formally accepted for publication once it meets all outstanding technical requirements.

Kind regards,

José María Gutiérrez

Academic Editor

PLOS ONE

Additional Editor Comments (optional):

Reviewers are satisfied with the changes and improvements introduced in the revised version of this manuscript.

Reviewers' comments:

Reviewer's Responses to Questions

**Comments to the Author**

1. If the authors have adequately addressed your comments raised in a previous round of review and you feel that this manuscript is now acceptable for publication, you may indicate that here to bypass the “Comments to the Author” section, enter your conflict of interest statement in the “Confidential to Editor” section, and submit your "Accept" recommendation.

Reviewer #1: All comments have been addressed

Reviewer #2: All comments have been addressed

2. Is the manuscript technically sound, and do the data support the conclusions?

Reviewer #1: Yes

Reviewer #2: Yes

3. Has the statistical analysis been performed appropriately and rigorously? 

Reviewer #1: Yes

Reviewer #2: Yes

4. Have the authors made all data underlying the findings in their manuscript fully available?

Reviewer #1: Yes

Reviewer #2: Yes

5. Is the manuscript presented in an intelligible fashion and written in standard English?

Reviewer #1: Yes

Reviewer #2: Yes

6. Review Comments to the Author

Reviewer #1: I recommend to accept as it is in this new version.

I recommend to accept as it is in this new version.

Reviewer #2: The revision of the article "Scorpion envenomation in the state of São Paulo, Brazil: Spatiotemporal analysis of a growing public health concern" by Lacerda et al. shows clear improvements. The remarks and suggestions I had made on the previous manuscript were all taken into account in the revised version. In my opinion, the paper can be published as is.

7. PLOS authors have the option to publish the peer review history of their article (what does this mean?). If published, this will include your full peer review and any attached files.

Reviewer #1: No

Reviewer #2: **Yes: **Jean-Philippe Chippaux

---

## [Editor Report · Acceptance letter]

31 Mar 2022

PONE-D-21-37124R1 

Scorpion envenomation in the state of São Paulo, Brazil: Spatiotemporal analysis of a growing public health concern 

Dear Dr. Lacerda:

I'm pleased to inform you that your manuscript has been deemed suitable for publication in PLOS ONE. Congratulations! Your manuscript is now with our production department. 

Kind regards, 

on behalf of

Dr. José María Gutiérrez 

Academic Editor

PLOS ONE